# Whole-Genome Analysis of Human Papillomavirus Type 16 Prevalent in Japanese Women with or without Cervical Lesions

**DOI:** 10.3390/v11040350

**Published:** 2019-04-16

**Authors:** Yusuke Hirose, Mamiko Onuki, Yuri Tenjimbayashi, Mayuko Yamaguchi-Naka, Seiichiro Mori, Nobutaka Tasaka, Toyomi Satoh, Tohru Morisada, Takashi Iwata, Tohru Kiyono, Takashi Mimura, Akihiko Sekizawa, Koji Matsumoto, Iwao Kukimoto

**Affiliations:** 1Department of Obstetrics and Gynecology, Showa University School of Medicine, Shinagawa-ku, Tokyo 142-8666, Japan; hysk1002@niid.go.jp (Y.H.); monuki@med.showa-u.ac.jp (M.O.); yuriten@med.showa-u.ac.jp (Y.T.); mayuneco@niid.go.jp (M.Y.-N.); m-bonby@med.showa-u.ac.jp (T.M.); sekizawa@med.showa-u.ac.jp (A.S.); matsumok@mui.biglobe.ne.jp (K.M.); 2Pathogen Genomics Center, National Institute of Infectious Diseases, Musashi-murayama, Tokyo 208-0011, Japan; moris@nih.go.jp; 3Department of Obstetrics and Gynecology, Faculty of Medicine, University of Tsukuba, Tsukuba, Ibaraki 305-8575, Japan; tsknbtk@gmail.com (N.T.); toyomi-s@md.tsukuba.ac.jp (T.S.); 4Department of Obstetrics and Gynecology, Keio University School of Medicine, Shinjuku-ku, Tokyo 160-0016, Japan; morisada@a7.keio.jp (T.M.); iwatatakashi@1995.jukuin.keio.ac.jp (T.I.); 5Division of Carcinogenesis and Cancer Prevention, National Cancer Center Research Institute, Chuo-ku, Tokyo 104-0045, Japan; tkiyono@ncc.go.jp

**Keywords:** papillomavirus, cervical cancer, HPV16, variant

## Abstract

Recent large-scale genomics studies of human papillomaviruses (HPVs) have shown a high level of genomic variability of HPV16, the most prevalent genotype in HPV-associated malignancies, and provided new insights into the biological and clinical relevance of its genetic variations in cervical cancer development. Here, we performed deep sequencing analyses of the viral genome to explore genetic variations of HPV16 that are prevalent in Japan. A total of 100 complete genome sequences of HPV16 were determined from cervical specimens collected from Japanese women with cervical intraepithelial neoplasia and invasive cervical cancer, or without cervical malignancies. Phylogenetic analyses revealed the variant distribution in the Japanese HPV16 isolates; overall, lineage A was the most prevalent (94.0%), in which sublineage A4 was dominant (52.0%), followed by sublineage A1 (21.0%). The relative risk of sublineage A4 for cervical cancer development was significantly higher compared to sublineages A1/A2/A3 (odds ratio = 6.72, 95% confidence interval = 1.78–28.9). Interestingly, a novel cluster of variants that branched from A1/A2/A3 was observed for the Japanese HPV16 isolates, indicating that unique HPV16 variants are prevalent among Japanese women.

## 1. Introduction

Genital human papillomaviruses (HPVs) are a group of sexually transmitted viruses, and are causatively linked with the development of cancers of the cervix, vulva, vagina, penis, anus, oropharynx, and oral cavity [1]. Among at least 13 genotypes (HPV16, -18, -31, -33, -35, -39, -45, -51, -52, -56, -58, -59, and -68) of high-risk HPVs, HPV16 is most frequently detected in cervical cancer, accounting for about 60% of invasive cervical cancer (ICC) cases worldwide [2]. HPV16 also predominates in other anogenital and oropharyngeal cancers [3], stressing the biological and clinical importance of HPV16 in HPV-associated malignancies. 

A different HPV type is defined by more than 10% differences in the *L1* capsid gene sequence from other types [4], and within each HPV type, there is further variability of the viral genomic sequence; 1–10% and 0.5–1% differences in complete viral genome sequence are classified as variant lineage and sublineage, respectively [5]. As such, HPV16 is composed of four lineages (A, B, C, and D), and at least nine sublineages (A1–4, B1–2, and D1–3) [5], and also harbors a large number of single nucleotide polymorphisms (SNPs) in each lineage/sublineage. Since HPV genome replication relies on high-fidelity DNA polymerases of the host cell, the sequence diversity of the HPV16 genome has presumably been generated through intracellular mutagenic processes, such as APOBEC-mediated viral genome editing and error-prone DNA repair [6,7], during the long history of human–HPV interaction [8].

Individual HPV16 variant lineages/sublineages display differential risk for cervical carcinogenesis. Among HPV16 variants, an increased risk of developing cervical intraepithelial neoplasia (CIN) 2/3 and ICC has been consistently reported for non-lineage A variants [9,10,11,12,13], in particular lineage D [14,15,16,17]. Regarding histologic subtypes of ICC, sublineages A1/A2/A3 variants (previously defined as “European” variants) were more prevalent in cervical squamous carcinoma, while lineage D variants were more frequently detected in cervical adenocarcinoma [18]. A recent large-scale genomics study of the HPV16 variant distribution has demonstrated that the A4 and D2/D3 sublineages are significantly associated with an increased risk of glandular lesions, and that the D2 sublineage shows the strongest increased risk of adenocarcinoma [19]. However, the risk assessment of HPV16 variants is complicated by the fact that the lineage/sublineage distributions vary between different countries and regions [20,21], and that the contribution of individual lineages/sublineages to cervical cancer development also differs by race and ethnicity [19,22].

In East Asian countries, including China, Japan, and South Korea, sublineage A4 variants were frequently detected in women with HPV16-positive cervical lesions [21,23,24,25], which previously led to their designation as “Asian” variants [26]. The A4 sublineage has been associated with a higher risk for the development of ICC compared to the European variants [23,24,27], whereas a recent study from China did not find a significant difference in the distribution of the A4 sublineage between CIN1/2/3 and ICC [28]. Intriguingly, in a large case/control study from the United States, Asian women infected with the A4 sublineage were at higher risk of developing CIN3 or worse compared with other racial groups [19], suggesting that this Asia-prevalent sublineage might confer a higher risk to these women once ethnicity is taken into account.

In the current study, we investigated the HPV16 lineage/sublineage distribution in Japan by determining the viral whole-genome sequences from Japanese women with CIN and ICC or without abnormal cervical cytology. Analysis of these data provided a risk association of the HPV16 variants, in particular the A4 sublineage, with cervical cancer development. 

## 2. Materials and Methods 

### 2.1. Study Samples

Cervical exfoliated cells were collected in ThinPrep media (Hologic, Bedford, MA, USA) using a cytobrush from Japanese patients diagnosed as negative for intraepithelial lesion or malignancy (NILM), CIN1, CIN2, CIN3, or ICC at Keio University Hospital, Tsukuba University Hospital, and Showa University Hospital from 2012 to 2017. The total cellular DNA was extracted from the recovered cells on a MagNA Pure LC 2.0 (Roche Diagnostic, Indianapolis, IN, USA), and subjected to PCR with PGMY09/11 primers to amplify HPV *L1* DNA, followed by reverse blot hybridization for HPV genotyping, as described previously [29]. The study protocol was approved by the Ethics Committees at each hospital and the National Institute of Infectious Diseases, and written informed consent for study participation was obtained from each participant.

### 2.2. Viral Whole-Genome Amplification and Next Generation Sequencing

Based on the genotyping results, DNA samples positive for HPV16 were subjected to long-range PCR to amplify the whole-genome sequences of HPV16, as described previously [30]. Full-circle PCR or overlapping PCR was performed using PrimeSTAR^®^ GXL DNA polymerase (Takara, Ohtsu, Japan) with the following primers: for full-circle PCR, HPV16-1742F (5′-TGC TGT CTA AAC TAT TAT GTG TGT CTC-3′) and HPV16-1873R (5′-GCG TGT CTC CAT ACA CTT CA-3′); for overlapping PCR, HPV16-1744F (5′-TGT CTA AAC TAT TAT GTG TGT CTC CAA TG-3′) and HPV16-5692R (5′-GAT ACT GGG ACA GGA GGC AAG TAG ACA GT-3′); HPV16-5531F (5′-GGG TCT CCA CAA TAT ACA ATT ATT GCT G-3′) and HPV16-1980R (5′-TAT CGT CTA CTA TGT CAT TAT CGT AGG CCC-3′). Among 238 HPV16-positive samples subjected to PCR, 100 samples successfully generated PCR products covering the complete viral genome. The amplified DNA separated using agarose gel electrophoresis was then purified with the Wizard gel purification kit (Promega, Madison, WI, USA). The purified DNA was converted to a short-fragmented DNA library using the Nextera XT DNA sample prep kit (Illumina, San Diego, CA, USA), followed by size selection with SPRIselect (Beckman Coulter, Brea, CA, USA). The multiplexed libraries were analyzed on a MiSeq sequencer (Illumina) with the MiSeq reagent kit v3 (150 cycle). The complete genome sequences of HPV16 were de novo assembled from the total read sequences using the VirusTAP pipeline [31] (https://gph.niid.go.jp/cgi-bin/virustap/index.cgi). The average depth of reads covering the HPV16 genome was approximately 11,000 per sample, and the minimum depth of 500 was used to confirm viral SNPs. The accuracy of the reconstructed whole-genome sequences was verified via read mapping with Burrows–Wheeler Aligner v0.7.12 [4] and subsequent visual inspection using Integrative Genomics Viewer v2.3.90 [32]. All HPV16 genome sequences determined were deposited to the DNA Data Bank of Japan (DDBJ) (accession numbers are shown in Appendix A), and the raw sequencing data are available from the DDBJ, Sequence Read Archive, under accession number DRA006584.

### 2.3. Phylogenetic Tree Construction

The complete genome sequences of HPV16 (*n* = 125), including 100 isolates determined in this study, 10 reference genomes that represent HPV16 variant (sub) lineages (A1, K02718; A2, AF536179; A3, HQ644236: A4, AF534061: B1, AF536180; B2, HQ644298; C, AF472509; D1, HQ644257; D2, AY686579; D3, AF402678), 14 isolates previously reported by us from Japanese women (AB818687, AB818688, AB818689, AB818690, AB818691, AB818692, AB818693, AB889488, AB889489, AB889490, AB889491, AB89492, AB889493, AB889494) [30], and one isolate reported from Thailand (FJ610151) [33], were aligned against each other using MAFFT v7.309 [34] with default parameters. Maximum likelihood trees were inferred using RAxML HPC v8.2.9 [35], under the general time-reversible nucleotide model with gamma-distributed rate heterogeneity and invariant sites (GTRGAMMAI), employing 1000 bootstrap values. Phylogenetic trees were visualized in FigTree v1.4.3 (http://tree.bio.ed.ac.uk/).

### 2.4. Sequencing of HPV16 E6/E7

The *E6/E7* sequence was amplified using PCR from the total cellular DNA with primers (HPV16-32F, 5′-GTA ACC GAA ATC GGT TGA ACC GAA ACC-3′, and HPV16-998R, 5′-CCT GTA TCA CTG TCA TTT TCG TTC TCG T-3′) using PrimeSTAR^®^ GXL DNA polymerase. The PCR condition consisted of 38 cycles of 98 °C for 10 s, 64 °C for 15 s, and 68 °C for 1 min. Amplification was verified using agarose gel electrophoresis, and successfully amplified DNA (approximately 950-bp length) was purified with the Wizard gel purification kit, followed by direct sequencing with the above-described PCR primers on a 3730 xl sequencer (Applied Biosystems, Austin, TX, USA).

### 2.5. Generation of Human Cervical Keratinocytes Expressing E7

The *E7* sequences of A1, A4, and A5 were amplified using PCR with primers (forward, 5′-TTG CGG CCG CAC CAT GCA TGG AGA TAC ACC TAC ATT GC-3′; and reverse, 5′-TTG CGG CCG CTG GTT TCT GAG AAC AGA TGG GGC ACA C-3′) from clinical specimens, and cloned into the *Not*I site of p3xFLAG-CMV14 (Sigma-Aldrich, St. Louis, MO, USA) to fuse the 3xFLAG sequence to the C-terminus of the E7 protein. Then, the E7-FLAG sequence was amplified using PCR, and cloned between the *Pac*I and *Xho*I sites of a retroviral transfer plasmid, pMXs-puro (Cell Biolabs, San Diego, CA, USA). A retrovirus vector expressing E7-FLAG was prepared via transfection of the transfer plasmid into GP2-293 packaging cells together with an envelope expression plasmid, pE-ampho (Takara). Human cervical keratinocytes immortalized with telomerase reverse transcriptase [36] were infected with the recombinant retrovirus expressing E7-FLAG, and selected with 1 μg/mL puromycin for 72 h, followed by culture without the drug for 48 h. The surviving cells were pooled and harvested for western blot analyses. 

### 2.6. Western Blotting

The total cell extract was prepared from the recovered cells by incubation in RIPA buffer (20 mM Tris-HCl (pH 8.0), 150 mM NaCl, 5 mM MgCl_2_, 1 mM EDTA, 1 mM dithiothreitol, 1% Nonidet P-40, and complete protease inhibitor cocktail (Roche Diagnostic)) at 4°C for 10 min, followed by centrifugation at 21,000× *g* at 4 °C for 10 min. Proteins in the extract were separated on a 4–15% polyacrylamide gel using SDS-PAGE, transferred to a PVDF membrane, and detected by the ECL prime western blotting detection reagent (GE Healthcare, Chicago, IL, USA) with the following antibodies: anti-FLAG (M2; Sigma-Aldrich), anti-pRb (4H1; Cell Signaling Technology, Danvers, MA), anti-PTPN14 (D5T6Y; Cell Signaling Technology), and anti-α-tubulin (B-5-1-2; Sigma-Aldrich). The band intensity of each protein was quantified with ChemiDoc XRS+ with Image Lab software (Bio-Rad, Hercules, CA, USA).

### 2.7. Statistical Analysis

A generalized linear model with binomial distribution and log link was used to calculate the odds ratio of progression from CIN1 to CIN2/3 or CIN2/3 to squamous cell carcinoma (SCC) for variant A sublineages with 95% confidence intervals (CI). The odds ratio was adjusted for the women’s age at the time of diagnosis. Fisher’s exact test was performed to evaluate a difference in sublineage distribution between different histological categories. A *p-*value < 0.05 was regarded as statistically significant. All statistical analyses were performed using R version 3.4.0 (https://cran.r-project.org/).

## 3. Results

### 3.1. Determination of HPV16 Whole-Genome Sequences from Japanese Women

To explore the prevalence of HPV16 variants in Japan, we collected cervical exfoliated cells from Japanese women with cervical lesions (CIN1, CIN2/3, and ICC) or without cervical abnormalities (NILM). Among HPV16-positive samples, a total of 100 samples (single infection, n = 82; multiple infections with other HPV types, n = 18) generated PCR products that covered the whole-genome sequence of HPV16. The number of women according to cervical disease status was as follows: NILM, n = 22; CIN1, n = 13; CIN2/3, n = 24; ICC, n = 41 (squamous cell carcinoma, n = 35; adenocarcinoma, n = 4; poorly differentiated carcinoma, n = 1; neuroendocrine carcinoma, n = 1). The average age (±standard deviation) in each disease category was as follows: NILM, 37.2 (±10.9); CIN1, 40.4 (±15.0); CIN2/3, 37.4 (±7.4); ICC, 42.1 (±12.1). 

We then performed next generation sequencing analyses of HPV16 DNA that had been amplified from the clinical samples. By using our established bioinformatics pipeline for reconstruction of HPV whole-genome sequences from short-read sequences [30], we successfully determined 100 complete HPV16 genome sequences (Appendix A). The length of the viral genome sequences ranged from 7863 to 7909 bp, with all isolates retaining the authentic HPV genomic organization comprised of six early genes (*E1*, *E2*, *E4*, *E5*, *E6*, and *E7*), two late genes (*L1* and *L2*), and two non-coding regions. Interestingly, nine isolates harbored insertion or deletion in *E1* or *E2/E4* as follows: five isolates with 60-, 63-, or 66-nucleotide insertions in *E1* (five unique isolates); two isolates with 11- or 33-nucleotide deletions in *E1*; and two isolates with 21- or 42-nucleotide deletions in *E2/E4*. The recovery of HPV16 whole-genome sequences suggests the presence of viral episomes in the corresponding clinical specimens, but we were unable to exclude a possibility of viral integration because a head-to-tail concatemeric HPV16 genome integrated into the host genome can also yield the viral whole-genome by PCR.

Out of the 100 women enrolled, 90 (90.0%) showed unique HPV16 isolates differing by at least one nucleotide from the sequences isolated from any other woman. Among the remaining 10 women, two identical pairs of isolates were shared by two women each (2 isolates, 4 women) and two identical isolates were shared by three women each (2 isolates, 6 women). Consequently, we obtained a total of 94 different HPV16 whole genome sequences from these Japanese women.

### 3.2. Phylogenetic Analysis of Japanese HPV16 Isolates

To reveal phylogenetic relationships between the Japanese HPV16 isolates, we constructed a whole-genome phylogenetic tree (Figure 1). We then assigned variant (sub) lineages to each isolate based on the variant reference genomes that were included in individual clusters. As summarized in Table 1, most of the Japanese isolates belonged to lineage A (*n* = 94, 94.0%), whereas two isolates (2.0%) belonged to lineage C and four isolates (4.0%) belonged to lineage D. No lineage B isolates were found among our study subjects. 

As shown in the phylogenetic tree, the 94 isolates of lineage A were grouped into two major clades including European and Asian variants, respectively. Consistent with a previous study from Japan [23], the prevalence of these two variants within the Japanese HPV16 isolates was similar: European (42 of 94, 44.7%) and Asian (52 of 94, 55.3%). Intriguingly, a small distinct cluster that branched from sublineages A1/A2/A3 near the root of the European clade was observed for the Japanese isolates, including two Japanese isolates previously reported by us (accession numbers, AB818687 and AB818688) [30]. The isolates in this cluster were most closely related to an HPV16 isolate previously reported from Thailand (accession number, FJ610151) [33] (Figure 1), but a similar cluster was not found in HPV16 isolates from the Netherlands (Appendix A) [37]. A more recent phylogenetic analysis of 211 complete HPV16 genome sequences also failed to detect the cluster [38]. These results suggest that this novel cluster exclusively contains Japanese HPV16 variants, which share a common ancestor with the Thai isolate. Since the complete genome sequences of these isolates and FJ610151 differ from the reference genome sequence of A1 (accession number, K02718) by an average 0.52% (*±*0.03% standard deviation), A2 (accession number, AF536179) by 0.49% (*±*0.02%), A3 (accession number, HQ644236) by 0.39% (*±*0.02%), and A4 (accession number, AF534061) by 0.70% (*±*0.02%), we hereafter refer to these variants as A5 variants.

### 3.3. Risk Assessment of HPV16 Variants for Cervical Cancer Development

Figure 2 shows the prevalence of HPV16 variant (sub) lineages according to the histological grades of cervical lesions. In NILM, CIN1, and ICC, A4 variants were predominantly detected among variant (sub) lineages, most notably in ICC (30 of 41, 73.2%), whereas A1 variants were most prevalent in CIN2/3 (12 of 24, 50.0%). Regarding A5 variants, their prevalence was highest in NILM (7 of 22, 31.8%), and gradually decreased across CIN1 (2 of 13, 15.4%), CIN2/3 (4 of 24, 16.7%), and ICC (3 of 41, 7.3%). 

The relative risk of cervical cancer development associated with individual sublineages was assessed by restricting the analyses for HPV16 single-infection cases in NILM, CIN, and squamous cell carcinoma (SCC). As shown in Table 2, the prevalence of the A4 sublineage was significantly higher in SCC (24 of 32, 75.0%) than in NILM/CIN1 (8 of 22, 36.4%) (Fisher’s exact test, p = 0.006) and CIN2/3 (7 of 22, 31.8%) (Fisher’s exact test, p = 0.002). Consistent with these observations, a higher risk of progression from CIN2/3 to SCC was observed for A4 compared to A1/A2/A3 (odds ratio = 6.72, 95% CI = 1.78–28.9). Although the prevalence of A5 was lower in SCC (2 of 32, 6.3%) than in NILM/CIN1 (4 of 22, 18.2%) and CIN2/3 (4 of 22, 18.2%), the difference was not statistically significant (Fisher’s exact test, p = 0.21). Accordingly, the relative risk of cervical lesion progression in individuals with A5 was not significantly different compared to the A1/A2/A3 sublineages (Table 2).

### 3.4. Genetic Variability among Sublineage A4 Variants

As the majority of the Japanese HPV16 isolates (n = 52) were grouped into the sublineage A4, we investigated variations in the whole-genome sequences of the A4 isolates. When compared them to the A4 reference genome sequence (accession number, AF534061), a total of 142 SNPs were detected across the genome of the Japanese isolates. As shown in Table 3, the highest variability was observed for the non-coding region between *E5* and *L2* (7.46%), followed by *E5* (3.57%) and the long control region (LCR) (2.64%), whereas the lowest variability was observed for *E1* (0.87%), followed by *L1* (1.25%) and *E6* (1.26%). Among 117 SNPs detected in the coding regions, 57 were synonymous substitutions and 60 were non-synonymous substitutions. *L2* harbored the largest number of non-synonymous substitutions (n = 21), followed by *E2* (n = 13) and *E5* (n = 8). In contrast, no non-synonymous substitutions were detected in the *E7* region.

### 3.5. Identification of SNPs Unique for A5 Variants

Multiple sequence alignment of the Japanese HPV16 isolates and reference HPV16 genomes revealed the presence of characteristic SNPs that distinguish A5 variants from other variant (sub) lineages. Table 4 shows these SNPs: C at nucleotide position 645, A at position 3068, C at position 4458, A at position 5042, and G at position 5836. Out of the five SNPs, three led to amino acid changes of the viral proteins as follows: Leu to Phe at amino acid position 28 in the E7 protein, Ala to Thr at position 105 in the E2 protein, and Ser/Pro/Ala to Asp/Asn at position 269 in the L2 protein.

### 3.6. HPV16 Variant Distribution in Cervical Adenocarcinoma

We were not able to amplify the viral whole-genome sequences from the majority of the HPV16-positive adenocarcinoma cases, possibly due to viral integration into the host genome. Thus, by utilizing viral SNPs in the *E6*/*E7* region, we further explored the HPV16 variant distribution among Japanese women with cervical adenocarcinoma. The *E6*/*E7* fragments were successfully amplified from 13 additional samples of adenocarcinoma with HPV16 single-infection and subjected to Sanger sequencing. Variant assignment was performed by reading the nucleotide of (sub) lineage-specific SNPs as follows: G at nucleotide position 145 for lineage A, G at position 647 for sublineage A4, and C at position 645 for A5 variants. Analysis of the 13 samples revealed that, as was the case in SCC, A4 variants were most prevalent in adenocarcinoma (9 of 13, 69.2%), followed by A1/A2/A3 variants (3 of 13, 23.1%) and A5 variants (1 of 13, 7.7%). As shown in Figure 3, when combined with four whole-genome isolates determined via next generation sequencing, the variant distribution in adenocarcinoma cases was as follows: A1/A2/A3 (3 of 17, 17.6%), A4 (11 of 17, 64.7%), A5 (2 of 17, 11.8%), and D2 (1 of 17, 5.9%). No significant difference was observed for the variant distribution between adenocarcinoma and SCC (Fisher’s exact test, p = 0.85).

### 3.7. Biological Activity of E7 Variants of Lineage A

The SNPs in the E7 region specific for A4 and A5 variants were non-synonymous substitutions compared to A1 and lead to amino acid changes in the E7 protein as follows: N29S for A4, and L28F for A5. Since these amino acid residues were in close proximity to the conserved LXCXE motif required for binding to the retinoblastoma protein (pRb) (Figure 4A), we examined whether these E7 variations affected its ability to degrade pRb. To this end, human cervical keratinocytes immortalized with telomerase reverse transcriptase were stably transduced with a retrovirus vector encoding FLAG-tagged E7 proteins of A1, A4, and A5, or an empty vector, and the total cell lysates were analyzed using western blotting. As shown in Figure 4B, similar levels of expression were observed for all three E7 variants. However, while the A1 and A4 variants decreased pRb levels to a similar extent, the A5 variant showed a slightly attenuated ability to degrade pRb, although the difference was not statistically significant. In contrast, all E7 variants similarly reduced levels of PTPN14, another tumor suppressor protein that is targeted by E7 [39,40], indicating that these amino acid residues were not involved in the degradation of PTPN14.

## 4. Discussion

This study reports for the first time the distribution of HPV16 variant lineages/sublineages in Japan based on the viral complete genome sequences obtained from 100 Japanese women with or without cervical lesions. Although many studies have so far relied on sequences of *E6*/*E7* and the LCR for assignment of HPV16 variant (sub) lineages, these regions lack appropriate diagnostic SNPs for some (sub) lineages [19]. Using the viral whole-genome sequence is thus the most reliable procedure to determine (sub) lineage classification for each HPV16 isolate. 

The overall distribution of HPV16 variants in Japan was strongly biased toward lineage A (94.0%), within which the frequency of European and Asian variants was similar (42.0% and 52.0%, respectively). This pattern of variant distribution is consistent with a previous study analyzing cervical specimens from CIN/ICC patients in Japan [41], which demonstrated the variant distribution as 54.4% for European variants and 44.9% for Asian variants [38]. On the other hand, a much higher prevalence of Asian (80.6%) versus European (19.4%) variants was reported in another study [42], which collected cervical swab samples from female sex workers in Japan; this may be due to demographic differences in sample source compared to our study. 

In agreement with a previous Chinese study including a meta-analysis of a relative risk of the A4 sublineage [24], we find a significantly higher risk of A4 for the development from high-grade CIN to ICC in Japanese women compared to A1/A2/A3. Intriguingly, a recent study in the United States demonstrated that a higher risk of A4 for CIN3 and cervical cancer was observed for Asian women compared to white/Hispanic women [19], which suggests that some matching of viral-host ethnicity plays an important role for this differential risk. Since HPV16 A4 variant genomes encode an E7 protein with a characteristic N29S substitution, we focused on whether this E7 substitution led to an enhancement of its major oncogenic activities, namely pRb- and PTPN14-degradation activities, thereby contributing to the higher risk of A4. However, there were no apparent differences in degradation activity between the E7 variants of A4 and A1, although the human cervical keratinocytes used in this study were derived from a Japanese woman and ethnically match the A4 sublineage. These results suggest that the viral gene(s) other than *E7* and/or the different activities of the LCR might be responsible for the higher risk of cervical cancer that is associated with A4 in Asian women. Interestingly, one A4-specific SNP in the LCR (C at nucleotide position 7176, in AF534061) generates a consensus binding site for the AP-1 transcription factor (TGANTCA), which may contribute to higher expression of *E6/E7* and thereby, higher oncogenic potential of A4. 

Although the D2 sublineage reportedly imposes a higher risk for adenocarcinoma compared to SCC [19], we did not observe a higher prevalence of D2 in adenocarcinoma cases among our Japanese HPV16 isolates. This discrepancy may be due to a low prevalence of D2 across all cervical histological grades in Japanese women, which precludes a risk assessment of this sublineage for predisposing HPV16-infected women to adenocarcinoma. Because A4 variants were predominantly detected in both adenocarcinoma and SCC over other variants in this study, the A4 sublineage may facilitate the development of ICC regardless of the histological types in which it is detected.

The E7 L28F substitution, which is characteristic for A5 variants, was previously identified in Japanese sex workers [42], and was also reported from Thailand [33] and northeast China [43]. A more recent study also described the same E7 substitution in one oropharyngeal cancer case in the United States [44]. To date, however, the complete genome sequence of a virus harboring the E7 L28F substitution has only been reported for one isolate in Thailand [33]. This study thus expands the number of whole-genome sequences of A5 variants, enabling detailed analyses of the genomic characteristics of these unique variants. 

Besides *E7*, the whole-genome sequences of A5 variants also contain particular non-synonymous substitutions in *E2* and *L2*. The *E2* substitution results in a change from alanine to threonine at position 105 of the E2 protein, which is located in its N-terminal transactivation domain [45]. We found that this substitution greatly reduced the intracellular level of the E2 protein in HeLa cells (data not shown), which is likely to have an impact on its biological functions. Furthermore, the substituted residue at position 269 of the L2 capsid protein lies in its chromatin-binding domain [46], thus potentially affecting the virus entry processes. We speculate that these amino acid changes may collectively lead to different biological behaviors of A5 variants compared to other lineage A variants. 

It is interesting to find some degree of prevalence for A5 variants among Japanese women (16 of 100 isolates, 16%). Intriguingly, although not statistically significant, we observed a slightly reduced ability of the E7 protein of A5 to target pRb for degradation. From an evolutionary perspective, the E7 L28F substitution might be unfavorable because it potentially weakens E7’s ability to promote the growth of HPV-infected cells through dysregulation of the pRb/E2F-pathways. In this regard, we note that this amino acid residue is located within one of the CD4+ T-cell epitopes of the HPV16 E7 protein [47]. Thus, the fitness cost inflicted by the E7 L28F substitution might be counterbalanced by immune evasion, such as an escape from recognition by helper T-cells, which may facilitate adaptation to the human host in order to maintain long-term persistent infection.

In conclusion, this study revealed the variant distribution of HPV16 in Japanese women, and reports that phylogenetically unique HPV16 variants are present in Japan. Since HPV16 genomic sequences exhibit much higher variability than previously thought [48], it is now critical to investigate the HPV16 genetic diversity in individual countries and regions, as was recently performed in Sweden [49]. Such viral genomics analyses will improve our understanding of the biological and epidemiological impact imposed by HPV genetic changes, and provide novel insight that can be leveraged in order to prevent or treat HPV-induced malignancies.

## Figures and Tables

**Figure 1 viruses-11-00350-f001:**
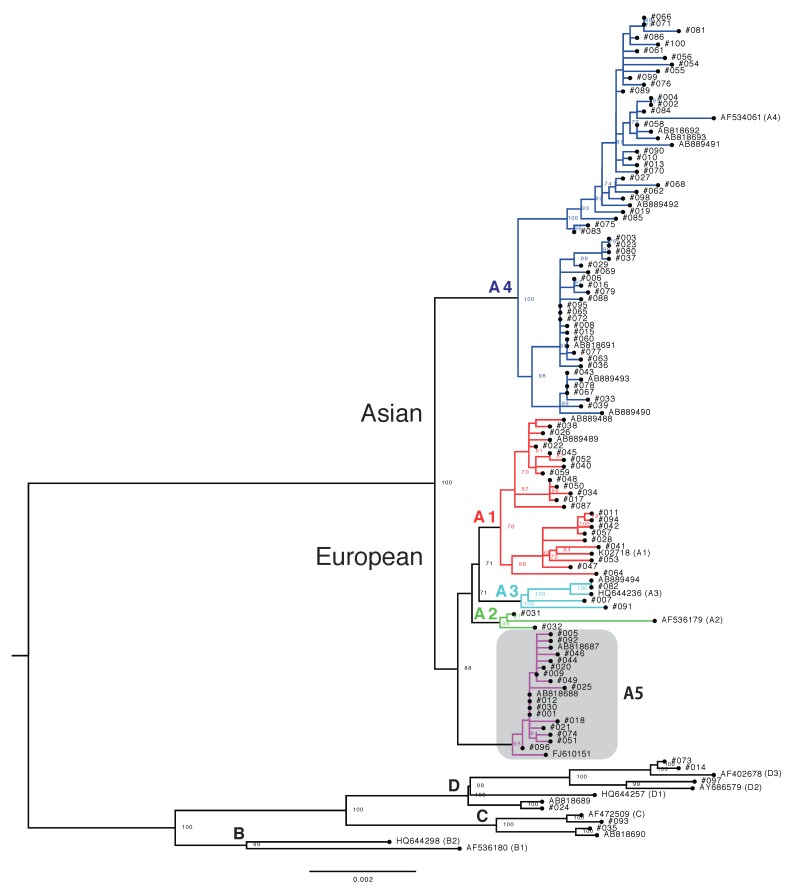
Maximum likelihood phylogenetic tree of Japanese HPV16 isolates. The phylogenetic tree was constructed using whole-genome sequences of 100 Japanese isolates obtained in this study, together with those of 14 Japanese isolates previously determined by us, one Thai isolate, and 10 reference strains for HPV16 variant (sub) lineages. Bootstrap values larger than 70% are displayed. Scale bar, nucleotide substitutions per site. A1, A2, A3, and A4: sublineages; B, C, and D: lineages. Gray area indicates a cluster of A5 variants.

**Figure 2 viruses-11-00350-f002:**
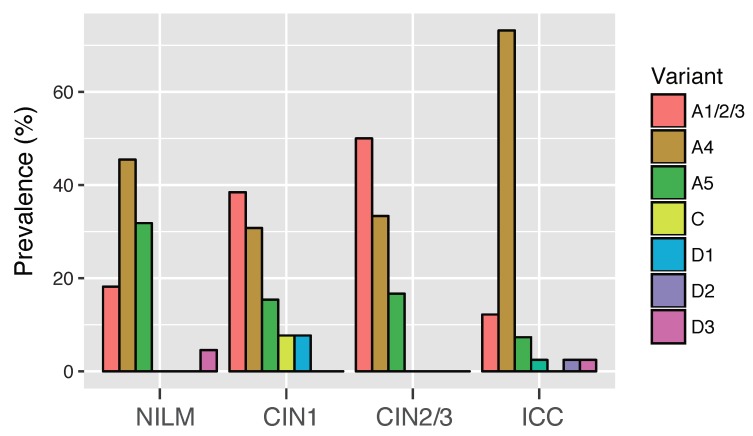
Distribution of HPV16 variant (sub) lineages across the histological grades of cervical lesions. The prevalence of each (sub) lineage is shown as the percentage in the total number of cases in each category. The number of samples in each category: NILM, *n* = 22; CIN1, *n* = 13; CIN2/3, *n* = 24; ICC, *n* = 41.

**Figure 3 viruses-11-00350-f003:**
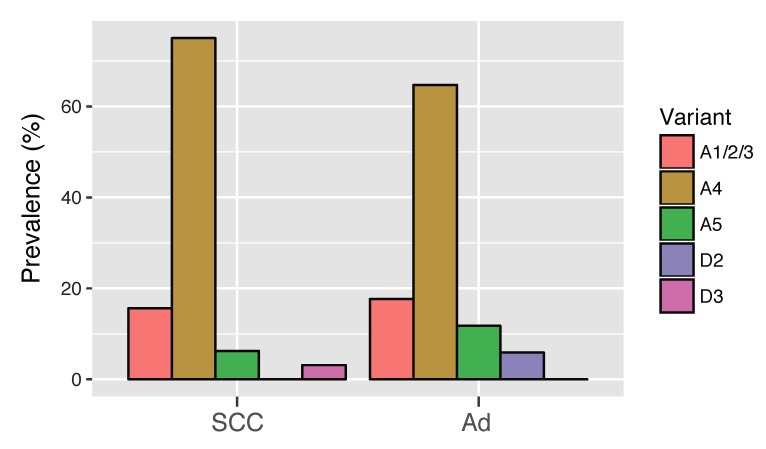
Distribution of HPV16 variant (sub) lineages in cervical squamous cell carcinoma and adenocarcinoma. HPV16 single-infection cases of squamous cell carcinoma (SCC) (*n* = 32) and adenocarcinoma (Ad) (*n* = 17) were classified into variant (sub) lineages. The prevalence of each (sub) lineage is shown as the percentage in the total number of cases in each category.

**Figure 4 viruses-11-00350-f004:**
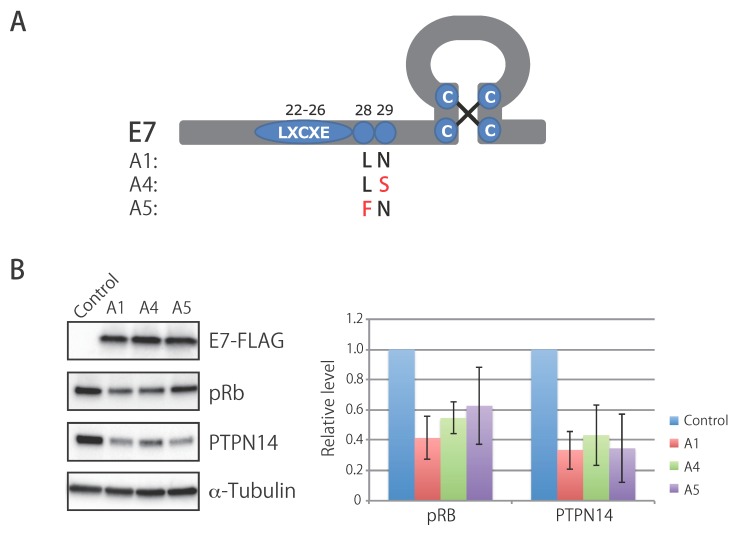
Biological activity of the E7 proteins of HPV16 lineage A variants. (**A**) Schematic representation of the HPV16 E7 protein. The LXCXE motif required for degradation of the retinoblastoma protein (pRb) and amino acid residues characteristic for A1, A4, and A5 are shown. (**B**) Western blot analyses of cell extracts from E7-transduced human cervical keratinocytes. Representative results of three independent experiments are shown in the left panel. Expression levels of pRb and PTPN14 were quantified as the relative band intensity of each protein adjusted with that of α-tubulin, and the average and standard deviation of the three experiments are shown in the right panel. No significant difference was observed for pRb levels between A1 and A5 (*p* = 0.23, paired *t*-test).

**Table 1 viruses-11-00350-t001:** Distribution of HPV16 variants according to cervical disease status (*n* = 100).

Variant	Total	NILM	CIN1	CIN2/3	ICC
All	100	22	13	24	41
A	94	21	11	24	38
A1	21	3	3	12	3
A2	2	0	2	0	0
A3	3	1	0	0	2
A4	52	10	4	8	30
A5	16	7	2	4	3
B	0	0	0	0	0
C	2	0	1	0	1
D	4	1	1	0	2
D1	1	0	1	0	0
D2	1	0	0	0	1
D3	2	1	0	0	1

NILM—negative for intraepithelial lesion or malignancy; CIN—cervical intraepithelial neoplasia; ICC—invasive cervical cancer.

**Table 2 viruses-11-00350-t002:** Distribution of HPV16 variants in single-infection cases (*n* = 76).

Variant	NILM/CIN1	CIN2/3	SCC	OR * (95% CI)	OR ** (95% CI)
All	22	22	32		
A1/2/3	8	11	5	1.00 (reference)	1.00 (reference)
A4	8	7	24	0.66 (0.16–2.62)	**6.72 (1.78–28.9)**
A5	4	4	2	0.77 (0.13–4.41)	0.07 (0.0003–1.83)
D1	1	0	0	ND	ND
D3	1	0	1	ND	ND

* CIN2/3 vs. NILM/CIN1; ** SCC vs. CIN2/3; NILM—negative for intraepithelial lesion or malignancy; CIN—cervical intraepithelial neoplasia; SCC—squamous cell carcinoma; OR—odds ratio; CI—confidence interval; ND—not determined. Statistically significant values are bolded.

**Table 3 viruses-11-00350-t003:** SNPs detected in Japanese A4 isolates (*n* = 52).

Region	Size (bp)	SNPs	Variable Sites (%) *	Synonymous	Non-Synonymous
All	7905 **	142	1.80	57	60
*E6*	477	6	1.26	3	3
*E7*	297	4	1.35	4	0
*E1*	1950	17	0.87	11	6
*E2*	1098	17	1.55	4	13
*E4*	288	7	2.43	4	3
*E5*	252	9	3.57	1	8
NC	134	10	7.46		
*L2*	1422	37	2.60	16	21
*L1*	1596	20	1.25	14	6
LCR	832	22	2.64		

* The percentage of variable sites in each region; ** the length of the sequence of AF534061; NC—non-coding region between *E5* and *L2*; LCR—long control region.

**Table 4 viruses-11-00350-t004:** SNPs characteristic for A5 variants.

Position *	A5	Others	Gene	Position **	A5	Others
645	C	A	*E7*	28	**Phe**	Leu
3068	A	G	*E2*	105	**Thr**	Ala
4458	C	A	*L2*	74	Pro	Pro
5042	A	C	*L2*	269	**Asp/Asn**	Ser/Pro/Ala
5836	G	A	*L1*	92	Ser	Ser

* Nucleotide position in AB818688; ** amino acid residue position in each viral protein. Amino acid residues different from those in other variants are bolded.

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
