# Peer review of "Whole-Genome Analysis of Human Papillomavirus Type 16 Prevalent in Japanese Women with or without Cervical Lesions"

_viruses, 2019, doi:10.3390/v11040350_

Reviewer 1 Report

This study examines in detail the sequence of HPV16 genomes in CIN and cancers from Japanese women. They show that 90% have unique HPV16 sequences of whole genomes and identify a A1-subgroup of HPV16 variants. They also show a significant link between progression and the A4 variants. They test the ability of E7 from A1, A4 and A1-sub to reduce Rb levels in immortalized cervical keratinocytes and show some biological variation.

This is a very well written paper and the data is well presented and sound analysis has been conducted. The findings are of significant interest to the field.

 Comments and questions

 1.    Line 16 alter the word ‘astonishingly’ to something a little more scientific

 2.    Line 95, 238 samples produced ‘whole’ HPV16 genomes. I have a few questions here: (1). Do these CIN and ICC therefore contain episomal forms of HPV16 or integrated concatemers? (2). Was the pathology for the ICC squamous and not glandular?

 3.    A4 isolates had a number of SNPs in non-coding regions including the LCR; are any of these changes in known transcription binding sites, E2 binding sites and origin of replication, or if between E5 and L2 are the changes near or in the polyA signal sequence?

 4.    Line 340 – make this clear that the non-synonymous changes in E7 of A4 and A1-sub is compared to A1.

 5.    Line 346, make clear that using Flag-tagged E7.

 6.    Figure 4, the decrease in the ability of A1-sub E7 to reduce RB levels is slight (29% reduced ability compared to 39-43%). The gel shown is one of three – what are the combined quantitation data?

Author Response

We thank the reviewers for their valuable comments and suggestions. Our responses to the comments are listed point-by-point below.

 Reviewer #1 (Comments for the Author):

This study examines in detail the sequence of HPV16 genomes in CIN and cancers from Japanese women. They show that 90% have unique HPV16 sequences of whole genomes and identify a A1-subgroup of HPV16 variants. They also show a significant link between progression and the A4 variants. They test the ability of E7 from A1, A4 and A1-sub to reduce Rb levels in immortalized cervical keratinocytes and show some biological variation.

 This is a very well written paper and the data is well presented and sound analysis has been conducted. The findings are of significant interest to the field.

 Comments and questions

 1.    Line 16 alter the word ‘astonishingly’ to something a little more scientific

Response

  As suggested by the reviewer, we have changed the word “astonishingly” to “extremely”. (line 16)

 2.    Line 95, 238 samples produced ‘whole’ HPV16 genomes. I have a few questions here: (1). Do these CIN and ICC therefore contain episomal forms of HPV16 or integrated concatemers? (2). Was the pathology for the ICC squamous and not glandular?

Response

  Since a head-to-tail concatemeric viral genome integrated in the host genome can also generate the full-length viral genome by PCR, we were not able to discriminate between episomal and integrated forms for our HPV16 isolates. This limitation has been described in the Results section as follows:

“The recovery of HPV16 whole-genome sequences suggests the presence of viral episomes in the corresponding clinical specimens, but we were unable to exclude a possibility of viral integration because a head-to-tail concatemeric HPV16 genome integrated into the host genome can also yield the viral whole-genome by PCR.” (lines 182-185)

   Most of the ICC samples generating whole HPV16 genomes were squamous cell carcinoma (35 out of 41), and it has been described in the Results section as follows:

“ICC, n = 41 (squamous cell carcinoma, n = 35; adenocarcinoma, n = 4; poorly differentiated carcinoma, n = 1; neuroendocrine carcinoma, n = 1)” (lines 169-170)

  3.    A4 isolates had a number of SNPs in non-coding regions including the LCR; are any of these changes in known transcription binding sites, E2 binding sites and origin of replication, or if between E5 and L2 are the changes near or in the polyA signal sequence?

Response

  According to the reviewer’s suggestions, we searched for A4-specific SNPs in the LCR and the non-coding region between E5 and L2. We found six A4-specific SNPs in the LCR, but none in the latter region. We then examined whether these SNPs overlap binding sequences for cellular or viral transcription factors in the LCR, and found that one SNP newly generates a consensus binding site for the AP-1 transcription factor. We have added this finding in the Discussion section as follows:

“Interestingly, an A4-specific SNP in the LCR (C at nucleotide position 7176, in AF534061) generates a consensus binding site for the AP-1 transcription factor (TGANTCA), which may contribute to higher expression of E6/E7 and thereby, higher oncogenic potential of A4.” (lines 406-409)

  4.    Line 340 – make this clear that the non-synonymous changes in E7 of A4 and A1-sub is compared to A1.

Response

  As suggested by the reviewer, we added the phrase “compared to A1” (line 345).

  5.    Line 346, make clear that using Flag-tagged E7.

Response

  As suggested, we described as “FLAG-tagged E7” (line 350).

  6.    Figure 4, the decrease in the ability of A1-sub E7 to reduce RB levels is slight (29% reduced ability compared to 39-43%). The gel shown is one of three – what are the combined quantitation data?

Response

  According to the reviewer’s suggestion, we have shown the combined quantitation data of three western blot experiments in Figure 4B. Although we see a trend of a slightly higher level of pRb with A1_sub (now called as A5) E7, the difference is subtle and not statistically significant between the E7 variants, so we soften our statement about the biological difference of A1-sub E7 throughout the manuscript. The corresponding statement has been deleted from the Abstract, and the Discussion section has been modified as follows:

“It is interesting to find some degree of prevalence for A5 variants among Japanese women (16 of 100 isolates, 16%). Intriguingly, although not statistically significant, we observed a slightly reduced ability of the E7 protein of A5 to target pRb for degradation. From an evolutionary perspective, the E7 L28F substitution might be unfavorable because it potentially weakens E7’s ability to promote the growth of HPV-infected cells through dysregulation of the pRb/E2F-pathways. In this regard, we note that this amino acid residue is located within one of the CD4+ T-cell epitopes of the HPV16 E7 protein [47]. Thus, the fitness cost inflicted by the E7 L28F substitution might be counterbalanced by immune evasion, such as an escape from recognition by helper T-cells, which may facilitate adaptation to the human host in order to maintain long-term persistent infection.” (lines 434-448)

  Finally, we noticed that in the previous manuscript one SCC sample (#056) was misclassified as CIN3. We corrected this mistake in the figures and tables, and the changes to the previous manuscript are highlighted by red text. We also re-calculated the odds ratios, and confirmed that the conclusion on variants risk was not affected by this correction. We apologize for the correction.

Reviewer 2 Report

Hirose and colleagues report the whole-genome analysis of 100 novel HPV16 genomes isolated from Japanese women. Interestingly, the authors identified a new phylogenetic cluster, indicating that there is still more HPV16 variability waiting to be described, further highlighting the need for these types of studies. I have some minor comments with regards to the bioinformatic analysis reported in the paper. However, the reported biochemical assay needs some additional controls and statistical analysis.

Major comments

1) The PCR based assay does not guarantee that only episomal copies are amplified (e.g., head to tail integrants would also be amplified). This should be discussed.

2) please include the accession numbers for the Japanese DNA databank. Please also provide a link to the raw read data.

3) It is not clear why the authors name the new phylogenetic clade A1-sub. The new clade is basal to A1, A2, and A3. Why not nake the clade A5. What is the pairwise sequence identity of the A1-sub clade compared to A2, A3, or A4? Finally, the 'Asian' and 'European' labels are confusing. If A1-sub is exclusively Japanese (as the authors claim, however, see below), there is no reason to call the A1, A1-sub, A2, and A3 clade European.

4) I am not sure that the authors should highlight "strikingly, no non-synonymous substitutions were detected in the E7 region." 

5) Figure 4 shows the ability of different E7 variants to degrade pRb and PTPN14. The authors argue that while there are no differences between the ability to degrade PTPN14, there are differences in pRb degradation. Based on the provided data, I disagree with this conclusion. The authors should quantify, plot, and provide statistical analysis for all three repeats, not just show a representative figure. Solely based on the presented data, the standard deviation of the remaining PTPN14 is larger than the SD of the remaining pRb. I am surprised by the relatively minimal degradation of pRb, I would have expected that most (if not all) pRb should be degraded. It may be worth adding MG132, and confirming proteasomal degradation of pRb in this system.

minor comments:

1) what is the average read coverage for each of the genomes. Specifically, how much support is there for the different SNPs?

2) What model was used during phylogenetic tree construction?

3) On line 173, the authors mention that there were four isolates with a 63- or 66 nt indel. Were these two samples from 2 patients, or were these fully unique sequences.

4) Why is clade C not included in table 2?

5) On line 400, the authors discuss that differences in E7's ability to degrade pRb or pTPN14 did not explain the observed epidemiological risk. However, this is not surprising. Even in viruses with larger risk association (e.g., HPV16 vs. HPV31) E7's ability to degrade E7 is not correlated with risk. This is not meant to be a criticism top the authors, but it is likely naive to think that a single protein-protein interaction would explain viral risk association.

Author Response

We thank the reviewers for their valuable comments and suggestions. Our responses to the comments are listed point-by-point below.

 Reviewer #2 (Comments for the Author):

Hirose and colleagues report the whole-genome analysis of 100 novel HPV16 genomes isolated from Japanese women. Interestingly, the authors identified a new phylogenetic cluster, indicating that there is still more HPV16 variability waiting to be described, further highlighting the need for these types of studies. I have some minor comments with regards to the bioinformatic analysis reported in the paper. However, the reported biochemical assay needs some additional controls and statistical analysis.

 Major comments

 1) The PCR based assay does not guarantee that only episomal copies are amplified (e.g., head to tail integrants would also be amplified). This should be discussed.

Response

  As suggested by the reviewer, we have discussed this issue as follows:

“The recovery of HPV16 whole-genome sequences suggests the presence of viral episomes in the corresponding clinical specimens, but we were unable to exclude a possibility of viral integration because a head-to-tail concatemeric HPV16 genome integrated into the host genome can also yield the viral whole-genome by PCR.” (lines 182-185)

 2) please include the accession numbers for the Japanese DNA databank. Please also provide a link to the raw read data.

Response

  The DDBJ accession numbers for our HPV16 isolates are described in Table S1. The raw sequencing data are also available from the DDBJ, and its accession number has been added to the Materials and Methods section.

 3) It is not clear why the authors name the new phylogenetic clade A1-sub. The new clade is basal to A1, A2, and A3. Why not nake the clade A5. What is the pairwise sequence identity of the A1-sub clade compared to A2, A3, or A4? Finally, the 'Asian' and 'European' labels are confusing. If A1-sub is exclusively Japanese (as the authors claim, however, see below), there is no reason to call the A1, A1-sub, A2, and A3 clade European.

Response

  We thank the reviewer for his/her insightful comments on our naming of the clade. According to the reviewer’s suggestion, we calculated the pairwise sequence identity between A1_sub and A1/2/3/4 genome sequences individually. The average sequence differences of A1_sub are 0.52% for A1, 0.49% for A2, 0.39% for A3, and 0.70% for A4 in the viral whole genome. Based on these data, we would like to call the new phylogenetic clade as A5. The Results section has been modified as follows:

“Since the complete genome sequences of these isolates differ from the reference genome sequences of A1 (accession number, K02718) by an average 0.52% (+ 0.02% standard deviation), A2 (accession number, AF536179) by an average 0.49% (+ 0.02%), A3 (accession number, HQ644236) by an average 0.39% (+ 0.02%), and A4 (accession number, AF534061) by an average 0.70% (+ 0.02%), we hereafter refer to these variants as A5 variants.” (lines 235-239)

   Regarding the European/Asian clades, this naming was originally introduced in the mid 1990s and has constantly been used for the phylogenetic classification of HPV16, so we would like to leave it in the manuscript for readers to easily grasp the identity of each clade. We also suppose that it is possible that future studies in European countries find similar HPV16 variants belonging to this new clade.

 4) I am not sure that the authors should highlight "strikingly, no non-synonymous substitutions were detected in the E7 region."

Response

  As suggested by the reviewer, we have changed the term “Strikingly” to “In contrast”. (line 294)

 5) Figure 4 shows the ability of different E7 variants to degrade pRb and PTPN14. The authors argue that while there are no differences between the ability to degrade PTPN14, there are differences in pRb degradation. Based on the provided data, I disagree with this conclusion. The authors should quantify, plot, and provide statistical analysis for all three repeats, not just show a representative figure. Solely based on the presented data, the standard deviation of the remaining PTPN14 is larger than the SD of the remaining pRb. I am surprised by the relatively minimal degradation of pRb, I would have expected that most (if not all) pRb should be degraded. It may be worth adding MG132, and confirming proteasomal degradation of pRb in this system.

Response

  According to the reviewer’s suggestion, we have shown the combined quantitation data of three western blot experiments in Figure 4B. As the reviewer pointed out, the difference in pRb levels is slight and not statistically significant between the E7 variants, so we soften our statement about the biological difference of A1-sub (now called as A5) E7 throughout the manuscript. The corresponding statement has been deleted from the Abstract, and the Discussion section has been modified as follows:

“It is interesting to find some degree of prevalence for A5 variants among Japanese women (16 of 100 isolates, 16%). Intriguingly, although not statistically significant, we observed a slightly reduced ability of the E7 protein of A5 to target pRb for degradation. From an evolutionary perspective, the E7 L28F substitution might be unfavorable because it potentially weakens E7’s ability to promote the growth of HPV-infected cells through dysregulation of the pRb/E2F-pathways. In this regard, we note that this amino acid residue is located within one of the CD4+ T-cell epitopes of the HPV16 E7 protein [47]. Thus, the fitness cost inflicted by the E7 L28F substitution might be counterbalanced by immune evasion, such as an escape from recognition by helper T-cells, which may facilitate adaptation to the human host in order to maintain long-term persistent infection.” (lines 434-448)

 minor comments:

 1) what is the average read coverage for each of the genomes. Specifically, how much support is there for the different SNPs?

Response

  The average depth of reads covering the HPV16 genome for each sample was approximately 11,000, and the minimum depth of 500 was used to confirm viral SNPs. This information has been added to the Materials and Methods section (lines 103-105).

 2) What model was used during phylogenetic tree construction?

Response

  We used the general time-reversible nucleotide model with gamma-distributed rate heterogeneity and invariant sites in RAxML, and it has been added to the Materials and Methods section as follows:

“Maximum likelihood trees were inferred using RAxML HPC v8.2.9 [35] under the general time-reversible nucleotide model with gamma-distributed rate heterogeneity and invariant sites (GTRGAMMAI), employing 1,000 bootstrap values.” (lines 118-121)

 3) On line 173, the authors mention that there were four isolates with a 63- or 66 nt indel. Were these two samples from 2 patients, or were these fully unique sequences.

Response

  These longer isolates are fully unique sequences isolated from different patients. We also corrected the total number of samples as follows:

“Interestingly, nine isolates harbored insertion or deletion in E1 or E2/E4 as follows: five isolates with 60-, 63- or 66-nucleotide insertions in E1 (five unique isolates)” (lines 179-180)

 4) Why is clade C not included in table 2?

Response

  For risk assessment, we restricted the analysis to HPV16 single-infection cases. Two cases with lineage C infection were co-infection cases with HPV56, as shown in Table S1, so we excluded these cases from the analysis and Table 2.

 5) On line 400, the authors discuss that differences in E7's ability to degrade pRb or pTPN14 did not explain the observed epidemiological risk. However, this is not surprising. Even in viruses with larger risk association (e.g., HPV16 vs. HPV31) E7's ability to degrade E7 is not correlated with risk. This is not meant to be a criticism top the authors, but it is likely naive to think that a single protein-protein interaction would explain viral risk association.

Response

  We have agreed with the reviewer’s comment. Besides E7, we have additionally discussed a potential involvement of LCR variations in the oncogenic potential of A4 as follows:

“Interestingly, an A4-specific SNP in the LCR (C at nucleotide position 7176, in AF534061) generates a consensus binding site for the AP-1 transcription factor (TGANTCA), which may contribute to higher expression of E6/E7 and thereby, higher oncogenic potential of A4.” (lines 406-409)

   Finally, we noticed that in the previous manuscript one SCC sample (#056) was misclassified as CIN3. We corrected this mistake in the figures and tables, and the changes to the previous manuscript are highlighted by red text. We also re-calculated the odds ratios, and confirmed that the conclusion on variants risk was not affected by this correction. We apologize for the correction.

Round  2

Reviewer 2 Report

The resubmitted manuscript by Hirose has addressed most of my earlier concerns. I have a couple of remaining comments

 On line 16. I disagree that HPV16 has extremely high levels of variability.

 Figure 1, why is FJ610151 not included in lineage A5?

 If I understand the data in table 2 correctly, A4 has an increased risk of SCC, but not CIN2/3 which is the precursor to SCC. This seems counter-intuitive? Could the authors explain this observation?

 In Figure 2 the authors use ICC, in table 2 SCC, are these used interchangeably?

Author Response

We thank you for your valuable comments on our manuscript.

The resubmitted manuscript by Hirose has addressed most of my earlier concerns. I have a couple of remaining comments

On line 16. I disagree that HPV16 has extremely high levels of variability.

Response

As suggested by the reviewer, we have removed “extremely”.

 Figure 1, why is FJ610151 not included in lineage A5?

Response

As suggested by the reviewer, we have included FJ610151 in A5.

 If I understand the data in table 2 correctly, A4 has an increased risk of SCC, but not CIN2/3 which is the precursor to SCC. This seems counter-intuitive? Could the authors explain this observation?

Response

In a recent report by Mirabello et al. (J Natl Cancer Inst, 108(9), 2016), none of the HPV16 variant lineages was associated with an increased risk of CIN2, but a higher risk of SCC was observed for A4, which is consistent with our results. Although the underlying mechanisms for the differential risk of A4 for SCC development are elusive, we speculate that particular cellular processes, such as invasion and metastasis, required for disease progression from CIN3 to SCC might be facilitated by A4 infection.

 In Figure 2 the authors use ICC, in table 2 SCC, are these used interchangeably?

Response

In Figure 2 we included all cases of invasive cervical cancer (ICC) including squamous cell carcinoma (SCC) and adenocarcinoma to show the overall distribution of HPV16 variants in ICC, whereas in Table 2 we only presented HPV16 single-infection cases of SCC for risk assessment.